# TopoFR: A Closer Look at Topology Alignment on Face Recognition

## Abstract

The field of face recognition (FR) has undergone significant advancements with the rise of deep learning. Recently, the success of unsupervised learning and graph neural networks has demonstrated the effectiveness of data structure information. Considering that the FR task can leverage large-scale training data, which intrinsically contains significant structure information, we aim to investigate how to encode such critical structure information into the latent space. As revealed from our observations, directly aligning the structure information between the input and latent spaces inevitably suffers from an overfitting problem, leading to a structure collapse phenomenon in the latent space. To address this problem, we propose TopoFR, a novel FR model that leverages a topological structure alignment strategy called PTSA and a hard sample mining strategy named SDE. Concretely, PTSA uses persistent homology to align the topological structures of the input and latent spaces, effectively preserving the structure information and improving the generalization performance of FR model. To mitigate the impact of hard samples on the latent space structure, SDE accurately identifies hard samples by automatically computing structure damage score (SDS) for each sample, and directs the model to prioritize optimizing these samples. Experimental results on several face benchmarks demonstrate the superiority of our TopoFR over the state-of-the-art methods. Code and models are available at: `https://anonymous.4open.science/r/TopoFR-82BB`.

## 1 Introduction

Face recognition (FR) is a critical biometric authentication technique that is widely applied in various applications, including electronic payments, smartphone lock screens, and video surveillance. In recent years, convolutional neural networks (CNNs) have achieved remarkable success in FR task, thanks to their powerful ability to autonomously extract face features from images. Existing studies on FR primarily focuses on constructing more discriminative face features by developing margin-based loss functions Deng et al. (2019); Wang et al. (2018b); Kim et al. (2022) and powerful network architectures Chang et al. (2020); Schroff et al. (2015); Zhong & Deng (2021). Recently, the success of unsupervised learning Wu et al. (2023a); Moor et al. (2020); Jiang et al. (2021); Lopes & Pedronette (2023) and graph neural networks Wong & Vong (2021); Wu et al. (2023b) has demonstrated the importance of data structure information in improving model generalization. However, to the best of our knowledge, how to effectively mine the potential structure information in large-scale face data has not investigated. Thus, in this paper, we extend our interests on building a cutting-edge FR framework through exploiting such powerful and substantial structure information.

First, we use Persistent homology (PH) Edelsbrunner et al. (2008); Barannikov (1994), a mathematical tool used in topological data analysis Carlsson (2009) to capture the underlying topological structure of complex point clouds, to investigate the evolution trend of structure information in existing FR framework and illustrate 3 observations: (1) as the amount of data increases, the topological structure of the input space becomes more and more complex, as verified in Figures 1(a)-1(d) (this observation is also consistent with the conclusions proposed in some theoretical works that conducted topological research on random points Bobrowski & Kahle (2018); Kahle (2011);(2) as the amount of data increases, the topological structure discrepancy (measured via topologically relevant distance Moor et al. (2020) between the persistence diagrams of two spaces) between the input space and the latent space becomes increasingly larger, as verified in Figure 2a;(3) The results in

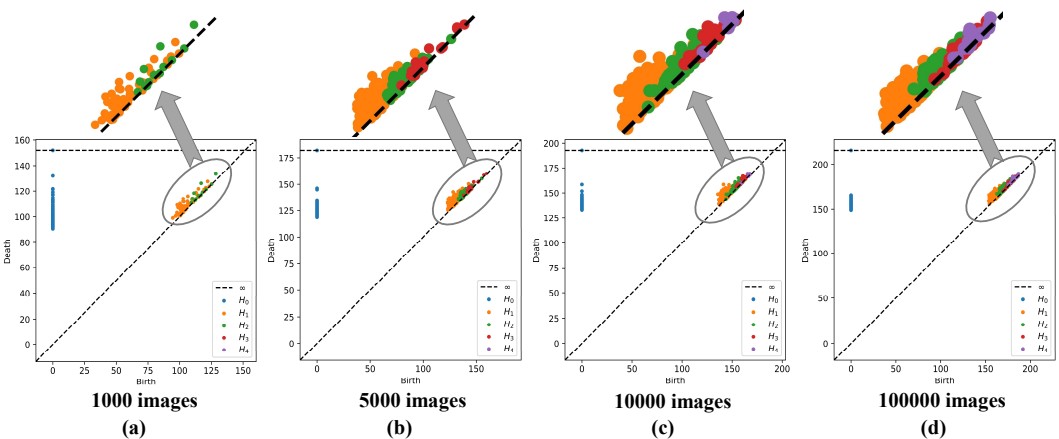

Figure 1: We sample 1000 (a), 5000 (b), 10000 (c) and 100000 (d) face images from the MS1MV2 dataset respectively, and compute their persistence diagrams using PH, where $H_j$ represents the $j$-th dimension homology. In topology theory, if the number of high-dimensional holes in the space is more, then the underlying topological structure of the space is more complex. Persistence diagram is a mathematical tool to describe the topological structure of space, where the $j$-th dimension homology $H_j$ in persistence diagram represents the $j$-th dimension hole in space. As shown in Figures 1(a)-1(d), as the amount of face data increases, the persistence diagram contains more and more high-dimensional homology (e.g., $H_3$ and $H_4$), which indicates that the input space contains an increasing number of high-dimensional holes. Therefore, this phenomenon also demonstrates a growing complexity in the topological structure of the input space.

Figure 2b demonstrate that as the depth of the network increases, the topological structure discrepancy becomes progressively smaller. This finding also provides an explanation for why models with more complex structure achieve higher FR accuracy. Based on the above observations, we can infer that in FR tasks with large-scale datasets, the structure of face data will be severely destroyed during training, which limits the generalization ability of FR models in practical application scenarios. To this end, we propose to improve the generalization performance of FR models by preserving the structure information.

However, we experimentally find that directly using PH to align the topological structures of the input space and the latent space may cause the model to suffer from **structure collapse phenomenon**. Concretely, under this experimental setting, we have 2 following quantitative and visualization results: (1) As shown in Figure 2c, the topological structure discrepancy drops to 0 dramatically during early training. (2) As illustrated in Figure 2d, when evaluating on the IJB-C benchmark Maze et al. (2018), there exists a significant structure information gap between the input space and the latent space. These typical overfitting phenomena indicates the latent space fails to preserve the structure information of input space accurately.

To remedy this issue, we propose a superior FR model named **TopoFR** that leverages a Perturbation-guided Topological Structure Alignment (**PTSA**) strategy to better preserve the topological structure information of the input space in corresponding latent face features. PTSA first employs a diverse data augmentation (**DDA**) mechanism to randomly augment training samples, effectively perturbing the latent space and increasing its structure diversity. Then PTSA utilizes an invariant structure alignment (**ISA**) mechanism to align the topological structures of the original input space and the perturbed latent space, resulting in face features with stronger generalization ability.

Moreover, in practical FR scenarios, the training dataset typically includes some low-quality face samples (i.e., hard samples) that are prone to being encoded into abnormal positions close to the decision boundary in the latent space Chang et al. (2020); Li et al. (2021b); Shi & Jain (2019), significantly destroying the topological structure of the latent space and affecting the alignment of structure. To address this issue, we propose a novel hard sample mining strategy named Structure Damage Estimation (**SDE**). SDE adaptively assigns structure damage score (**SDS**) to each sample based on its prediction uncertainty and prediction probability. By prioritizing the optimization of

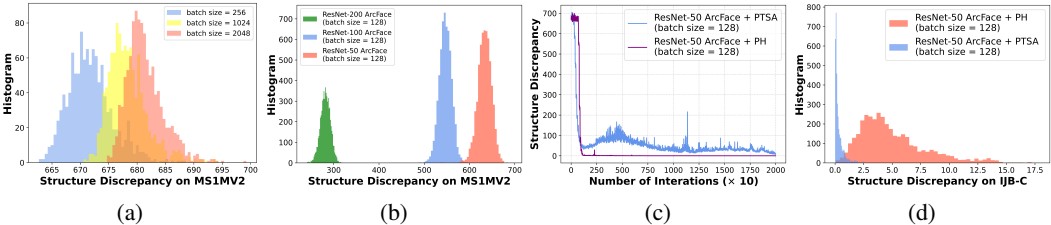

Figure 2: (a): We investigate the relationship between the amount of data and the topological structure discrepancy by employing ArcFace Deng et al. (2019) model (based on ResNet-50) to perform inferences on MS1MV2 training set. Inferences are conducted for 1000 iterations with batch sizes of 256, 1024, and 2048, respectively. Histograms are used to approximate these discrepancy distributions. (b): We investigate the relationship between the network depth and the topological structure discrepancy by performing inference on MS1MV2 training set (batch size=128) using ArcFace models with different backbones. (c): We investigate the trend of topological structure discrepancy during training (batch size=128) and found that i) directly using PH to align the topological structures will cause the discrepancy to drops to 0 dramatically; ii) whereas using our PTSA strategy promotes a smooth convergence of structure discrepancy. (d): Aligning the topological structures directly using PH (batch size=128) will lead to significant discrepancy when evaluating on IJB-C benchmark. Our PTSA strategy effectively mitigates this overfitting issue, resulting in smaller structure discrepancy during evaluation.

hard samples with significant structure damage, SDE can gradually guide these samples back to their reasonable positions, thereby improving the generalization ability of FR model.

In summary, the main contributions are listed as follows:

1) To the best of knowledge, we are the first to explore the topological structure alignment in FR task. We propose a novel topological structure alignment strategy called PTSA to effectively align the structures of the original input space and the perturbed latent space.

2) A novel hard sample mining strategy named SDE is introduced to mitigate the adverse impact of hard samples on the latent space structure.

3) Our experimental results show that the proposed method outperforms state-of-the-art methods on several face benchmark datasets, e.g., by employing ResNet-100 as the backbone, we achieve an accuracy of 96.95% on "TAR@FAR=1e-4" of the IJB-C benchmark using MS1MV2 training set.

## 2 RELATED WORKS

**Face Recognition (FR).** CNNs have shown remarkable progress in extracting face features for FR, using two primary methods: metric learning-based and margin-based softmax approaches. The former utilizes loss functions like Triplet loss Schroff et al. (2015), Tuplet loss Sohn (2016), and Center loss Wen et al. (2016) to learn discriminative face features, while the latter aims to incorporate margin penalty into the softmax loss framework, including methods such as ArcFace Deng et al. (2019), CosFace Wang et al. (2018b), AM-softmax Wang et al. (2018a), and SphereFace Liu et al. (2017). Recent studies have explored various techniques, including adaptive parameters Kim et al. (2022); Meng et al. (2021), mining Huang et al. (2020); Xu et al. (2021); Deng et al. (2020), learning acceleration An et al. (2021); Li et al. (2021a); An et al. (2022), and data uncertainty Li et al. (2021b); Chang et al. (2020); Shi & Jain (2019) to further enhance the efficiency of margin-based softmax loss on large-scale datasets. Moreover, the recent proposed Face transformer Zhong & Deng (2021) has demonstrated the potential of using Vision Transformer in FR.

**Persistent Homology (PH).** Over the past decade, PH has demonstrated significant advantages in multiple various such as signal processingPerea & Harer (2015); Guillemard & Iske (2011), video analysis Tralie & Perea (2018), neuroscience Singh et al. (2008); Dabaghian et al. (2012), disease diagnosis Chung et al. (2018) and evaluation of embedding strategies Rieck & Leitte (2015; 2017). In the field of machine learning, some studies Khramtsova et al. (2022); Venkataraman et al. (2016)

have demonstrated that integrating topological representations into neural network can enhance the model's recognition performance. Hofer et al. (2017) introduces a novel network layer that learns to project persistence diagrams into feature descriptors, and Som et al. (2020) develops a differentiable structure to accelerate the calculation of persistence images for input data. Moreover, Moor et al. (2020) derives a topological loss term to coordinate the structure of the input space and hidden space in autoencoder. Refs. Wong & Vong (2021) and Clough et al. (2020) introduce several PH-related loss functions to improve the network's segmentation performance. Horak et al. (2021) proposes a topology distance for the evaluation of GANs.

## 3 BACKGROUND: PERSISTENT HOMOLOGY

PH is a computational topology method that quantifies the changes in the topological invariants of a Vietoris-Rips complex as a scale parameter $\rho$ is varied. In this section, we briefly introduce some key concepts of PH. Further details on PH can be found in Refs. Edelsbrunner et al. (2008; 2000).

**Notation.** $\mathcal{X} := \{x_i\}_{i=1}^n$ represents a point cloud and $\mu : \mathcal{X} \times \mathcal{X} \to \mathbb{R}$ denotes a distance metric over $\mathcal{X}$. Matrix $\mathcal{M}$ represents the pairwise distances (i.e., Euclidean distance) between points in $\mathcal{X}$.

**Vietoris-Rips Complex.** The Vietoris-Rips complex Vietoris (1927) is a special simplicial complex constructed from a set of points in a metric space, and it can be used to approximate the topology of the underlying space. For $0 \le \rho < \infty$, we represent the Vietoris-Rips complex of point cloud $\mathcal{X}$ at scale $\rho$ as $\mathcal{V}_\rho(\mathcal{X})$, which contains all simplices (i.e., subsets) of $\mathcal{X}$, and each component of point cloud $\mathcal{X}$ satisfies a distance constraint: $\mu(x_i, x_j) \le \rho$ for any $i, j$. Moreover, the Vietoris-Rips complex satisfies a nesting relation, i.e., $\mathcal{V}_{\rho_i} \subseteq \mathcal{V}_{\rho_j}$ for any $\rho_i \le \rho_j$, which allows us to track the evolution progress of simplical complex as the scale $\rho$ increases. It is worth noting that $\mathcal{V}_\rho(\mathcal{X})$ and $\mathcal{V}_\rho(\mathcal{M})$ are equivalent because constructing the Vietoris-Rips complex only requires distance.

**Homology Group.** The homology group is an algebraic structures that analyzes the topological features of a simplicial complex in different dimension $j$, such as connected components ($H_0$), cycles/loops ($H_1$), voids/cavities ($H_2$), and higher-dimensional features ($H_j, j \ge 3$). By tracking the changes in topological features ($H_j$) of the Vietoris-Rips complex as the scale $\rho$ increases, it is possible to gain insight into the multi-scale topological information of the underlying space.

**Persistence Diagram and Persistence Pairing.** The persistence diagram $\mathcal{D}$ is a multi-set of points $(b, d)$ in the Cartesian plane $\mathbb{R}^2$, which encodes information about the lifespan of topological features. Specially, it summarizes the birth time $b$ and death time $d$ information of each topological feature in a homology group, where birth time $b$ signifies the scale at which the feature is created and death time $d$ refers to the scale at which it is destroyed. The difference between the death and the birth times is the lifetime of the homology group $l_f = |d - b|$. The persistence pairing $\gamma$ contains indices $(i, j)$ corresponding to simplices $r_i, r_j \in \mathcal{V}_\rho(\mathcal{X})$ that create and destroy the topological features identified by $(b, d) \in \mathcal{D}$, respectively.

## 4 METHODOLOGY

In this paper, we propose a novel framework named TopoFR for constraining the FR model to preserve the topological structure information of the input space in their latent features. The architecture of our TopoFR model is depicted in Figure 3. It consists of two components: a feature extractor $\mathcal{F}$ and an image classifier $\mathcal{C}$. Mathematically, given an input image $x$, the latent feature extracted by $\mathcal{F}$ is denoted as $f = \mathcal{F}(x) \in \mathbb{R}^l$, and the classification probability predicted by $\mathcal{C}$ is denoted as $g = \mathcal{C}(f) \in \mathbb{R}^K$, where $l$ represents the feature dimension and $K$ denotes the number of classes. The entropy of the classification prediction probability $g$ can be represented as $E(g) = -\sum_{k=1}^K g^k \log g^k$, where $g^k$ is the probability of predicting a sample to class $k$.

### 4.1 PERTURBATION-GUIDED TOPOLOGICAL STRUCTURE ALIGNMENT

As mentioned in Section 1, directly applying PH to align the topological structures of the input space and the latent space can cause the FR model to encounter structure collapse phenomenon. To remedy this problem, we propose a Perturbation-guided Topological Structure Alignment (**PTSA**) strategy that includes two mechanisms: diverse data augmentation and invariant structure alignment.

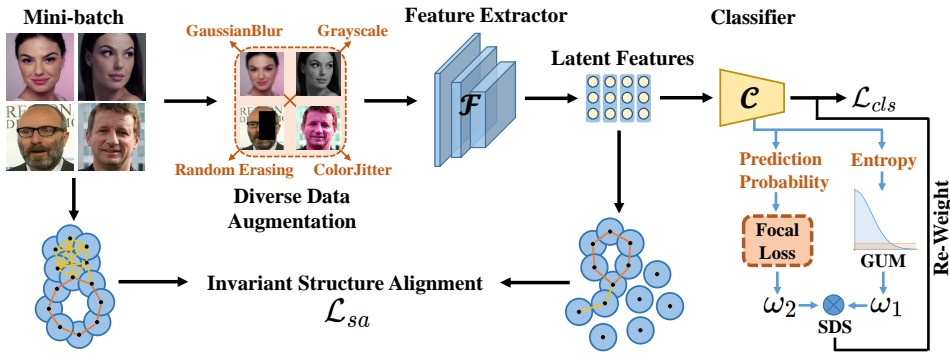

Figure 3: Global overview of our proposed TopoFR model. To effectively preserve the structure information of face data, PTSA strategy first employs DDA mechanism to augment training samples, and then utilizes ISA mechanism to align the topological structures of the original input space and the perturbed latent space. Moreover, to alleviate the damage of hard samples to the latent space structure, SDE strategy adaptively computes SDS for each sample and encourages the model to focus on optimizing hard samples with high SDS. $\otimes$ represents the multiplication operation.

**Diverse Data Augmentation (DDA).** PTSA first utilzes the DDA mechanism to randomly augment training samples. Specially, it introduces a data augmentation list $\mathcal{A} = \{\mathcal{A}_1, \mathcal{A}_2, \mathcal{A}_3, \mathcal{A}_4\}$ that includes four common data augmentation operations, namely Random Erasing $\mathcal{A}_1$ Zhong et al. (2020), GaussianBlur $\mathcal{A}_2$, Grayscale $\mathcal{A}_3$ and ColorJitter $\mathcal{A}_4$. For each training sample $x_i$, DDA will randomly select an operation $\mathcal{A}_r$ from $\mathcal{A}$ to augment it, i.e., $\widetilde{x}_i = \mathcal{A}_r(x_i)$. Then the augmented sample $\widetilde{x}_i$ will be fed into the model for supervised learning, which effectively increases the structure diversity of the latent space. In our model, we adopt the most widely used ArcFace loss Deng et al. (2019) as the basic classification loss:

$$\mathcal{L}_{arc}(\widetilde{x}_i, y_i) = -\log \frac{e^{s(\cos(\theta_i^y + m))}}{e^{s(\cos(\theta_i^y + m))} + \sum_{k=1, k \neq y}^{K} e^{s \cos \theta_i^k}}, \quad (1)$$

where $y_i$ indicates the class label of the original image $x_i$, $s$ is a scaling hyperparameter, $\theta_i^k$ is the angle between the $k$-th class center and feature, and $m > 0$ denotes an additive angular margin.

**Invariant Structure Alignment (ISA).** Given a mini-batch of original training samples $\mathcal{X} = \{x_i\}_{i=1}^n$, we denote the augmented batch samples as $\widetilde{\mathcal{X}} = \{\widetilde{x}_i\}_{i=1}^n$. For the augmented samples, we denote the latent features extracted by $\mathcal{F}$ as $\widetilde{\mathcal{Z}} = \left\{\widetilde{f}_i\right\}_{i=1}^n$. During forward propagation, we can construct the Vietoris-Rips complexes $\mathcal{V}_\rho(\mathcal{X})$ and $\mathcal{V}_\rho(\widetilde{\mathcal{Z}})$ for point clouds $\mathcal{X}$ and $\widetilde{\mathcal{Z}}$ respectively, based on their respective pairwise distance matrix (i.e., Euclidean distance) $\mathcal{M}^{\mathcal{X}}$ and $\mathcal{M}^{\widetilde{\mathcal{Z}}}$. Then we can compute the persistent homology of $\mathcal{V}_\rho(\mathcal{X})$ and $\mathcal{V}_\rho(\widetilde{\mathcal{Z}})$, and obtain their corresponding persistence diagrams $\left\{\mathcal{D}^{\mathcal{X}}, \mathcal{D}^{\widetilde{\mathcal{Z}}}\right\}$ and persistence pairings $\left\{\gamma^{\mathcal{X}}, \gamma^{\widetilde{\mathcal{Z}}}\right\}$, respectively.

Ideally, no matter how the face image is perturbed, the position of the encoded face feature in the latent space should remain unchanged, and the topological structure of the perturbed latent space should also be consistent with the original input space. To this end, we choose to align the original input space $\mathcal{X}$ with the perturbed latent space $\widetilde{\mathcal{Z}}$ to achieve this goal. To align the topological structure of two spaces, prior studies usually utilize bottleneck distance or p-Wasserstein distance to measure the discrepancy Turner et al. (2014); Mileyko et al. (2011) between persistence diagrams. However, these two distance metrics are sensitive to outliers Horak et al. (2021); Bertsekas (1981) and will increase the training time of the FR models, rendering them unsuitable for FR tasks with extremely large-scale datasets.

To mitigate this issue, we turn to retrieve the persistence diagrams values by subsetting the corresponding pairwise distance matrix with edge indices provided by the persistence pairings Zomorodian (2010); Moor et al. (2020), i.e., $\mathcal{D}^{\mathcal{X}} \simeq \mathcal{M}^{\mathcal{X}}[\gamma^{\mathcal{X}}]$ and $\mathcal{D}^{\mathcal{Z}} \simeq \mathcal{M}^{\widetilde{\mathcal{Z}}}[\gamma^{\widetilde{\mathcal{Z}}}]$. By comparing the difference between two topologically relevant distance matrices from both spaces, we can quickly

and stably compute the discrepancy between their persistence diagrams, providing an efficient solution for structure alignment of FR models driven by large-scale datasets. We formulate the ISA loss as follows:

$$\mathcal{L}_{sa}(\mathcal{D}^{\mathcal{X}}, \mathcal{D}^{\widetilde{\mathcal{Z}}}) = \frac{1}{2}\left(\left\|\mathcal{M}^{\mathcal{X}}[\gamma^{\mathcal{X}}] - \mathcal{M}^{\widetilde{\mathcal{Z}}}[\gamma^{\mathcal{X}}]\right\|^2 + \left\|\mathcal{M}^{\widetilde{\mathcal{Z}}}[\gamma^{\widetilde{\mathcal{Z}}}] - \mathcal{M}^{\mathcal{X}}[\gamma^{\widetilde{\mathcal{Z}}}]\right\|^2\right) \tag{2}$$

## 4.2 STRUCTURE DAMAGE ESTIMATION

In practical FR scenarios, low-quality face samples, also known as "hard samples", are commonly included in the training set. These hard samples tend to be encoded in abnormal positions near the decision boundary in the latent space Chang et al. (2020); Li et al. (2021b); Shi & Jain (2019), which will disrupt the latent space's topological structure and hinder the alignment of structure. To address this issue, we propose a novel hard sample mining strategy called Structure Damage Estimation (**SDE**). SDE is specifically designed to identify hard samples with serious structure damage within the training set accurately. By prioritizing the learning of these hard samples and guiding them back to the reasonable positions during optimization, SDE aims to mitigate the adverse impact of hard samples on the topological structure of the latent space.

**Prediction Uncertainty.** Hard samples are typically distributed near the decision boundary, thus have a high prediction uncertainty (i.e., large entropy of the classifier prediction) and are more likely to be misclassified by the classifier. Conversely, easy samples are usually located far from the decision boundary and have relatively low prediction uncertainty. To model the difficulty of each sample, we introduce a binary random variable $u_i \in \{0, 1\}$ for each sample $\widetilde{x}_i$ to indicate whether the sample is hard or easy by values of 1 and 0, respectively. Then the probability that sample $\widetilde{x}_i$ belongs to hard samples (i.e., with large prediction uncertainty) can be defined as $h_\varphi(\widetilde{x}_i) = P_\varphi(u_i = 1|\widetilde{x}_i)$, where $\varphi$ represents the parameter set. According to the cluster assumption Chapelle & Zien (2005), we believe that samples with higher prediction entropy are more disruptive to the topological structure of the latent space. Therefore, we propose to model the distribution of the entropy $E(\widetilde{g}_i)$ for each training sample $\widetilde{x}_i$ using the Gaussian-uniform mixture (**GUM**) model, a statistical distribution that is robust to outliers De Angelis et al. (2015); Lathuilière et al. (2018):

$$p\left(E(\widetilde{g}_i)|\widetilde{x}_i\right) = \pi\mathcal{N}^+(E(\widetilde{g}_i)|0, \Sigma) + (1 - \pi)\mathcal{U}(0, \Omega), \tag{3}$$

where

$$\mathcal{N}^+(a|0, \Sigma) = \begin{cases} 2\mathcal{N}(a|0, \Sigma), & a \geq 0. \\ 0, & a < 0. \end{cases} \tag{4}$$

$\mathcal{U}(0, \Omega)$ is a uniform distribution defined on $[0, \Omega]$, $\pi$ is a prior probability, and $\Sigma$ is the variance of Gaussian distribution $\mathcal{N}(a|0, \Sigma)$. In this mixed model, the uniform distribution term $\mathcal{U}$ and the Gaussian distribution term $\mathcal{N}^+$ respectively model the hard samples and easy samples. Then the posterior probability that the sample $\widetilde{x}_i$ to be hard (i.e., high-uncertainty) can be computed as follows:

$$h_\varphi(\widetilde{x}_i) = P_\varphi(u_i = 1|\widetilde{x}_i) = \frac{(1 - \pi)\mathcal{U}(0, \Omega)}{\pi\mathcal{N}^+(E(\widetilde{g}_i)|0, \Sigma) + (1 - \pi)\mathcal{U}(0, \Omega)}. \tag{5}$$

In Equation (5), when the classifier prediction probability is close to uniform distribution, the posterior probability of a sample belonging to hard data will be very high, i.e., $(\widetilde{g}_i \to [\frac{1}{K}, \frac{1}{K}, \cdots, \frac{1}{K}], h_\varphi(\widetilde{x}_i) \to 1)$, otherwise it is very low. By this mean, the prediction uncertainty of sample $x_i$ can be measured by a quantitative probability $h_\varphi(\widetilde{x}_i)$.

Assume $\widehat{E}(\widetilde{g}_i) = (-1)^{\epsilon_i}E(\widetilde{g}_i), \epsilon_i \sim B(1, 0.5)$, where $B$ is a Bernoulli distribution Dai et al. (2013), then variable $\widehat{E}(\widetilde{g}_i)$ obeys the following statistical distribution:

$$p\left(\widehat{E}(\widetilde{g}_i)|x_i\right) = \pi\mathcal{N}(\widehat{E}(\widetilde{g}_i)|0, \Sigma) + (1 - \pi)\mathcal{U}(-\Omega, \Omega). \tag{6}$$

In this way, the parameter set $\varphi = \{\pi, \Sigma, \Omega\}$ of GUM can be estimated via the Expectation-Maximization (EM) algorithm Coretto & Hennig (2016) with the following iterative formulas:

$$h_\varphi^{(t+1)}(\widetilde{x}_i) = \frac{(1 - \pi^{(t)})\mathcal{U}(-\Omega^{(t)}, \Omega^{(t)})}{\pi^{(t)}\mathcal{N}(\widehat{E}(\widetilde{g}_i)|0, \Sigma^{(t)}) + (1 - \pi^{(t)})\mathcal{U}(-\Omega^{(t)}, \Omega^{(t)})}, \pi^{(t+1)} = \frac{\sum_{i=1}^n(1 - h_\varphi^{(t+1)}(\widetilde{x}_i))}{n},$$

$$\Sigma^{(t+1)} = \frac{\sum_{i=1}^{n}(1 - h_\varphi^{(t+1)}(\widetilde{x}_i))(\widehat{E}(\widetilde{g}_i))^2}{\sum_{i=1}^{n}(1 - h_\varphi^{(t+1)}(\widetilde{x}_i))}, \Omega^{(t+1)} = \sqrt{3(\eta_2 - \eta_1^2)}, \tag{7}$$

where

$$\eta_1 = \frac{\sum_{i=1}^{n} \frac{h_\varphi^{(t+1)}(\widetilde{x}_i)}{1 - \pi^{(t+1)}} \widehat{E}(\widetilde{g}_i)}{\sum_{i=1}^{n}(1 - h_\varphi^{(t+1)}(\widetilde{x}_i))}, \eta_2 = \frac{\sum_{i=1}^{n} \frac{h_\varphi^{(t+1)}(\widetilde{x}_i)}{1 - \pi^{(t+1)}}(\widehat{E}(\widetilde{g}_i))^2}{\sum_{i=1}^{n}(1 - h_\varphi^{(t+1)}(\widetilde{x}_i))}.$$

**Structure Damage Score (SDS).** Compared to correctly classified samples, misclassified samples usually have larger difficulty and have greater destructive effects on the latent space's topological structure. Therefore, misclassified samples need to receive more attention from the model during training. Inspired by the Focal loss Lin et al. (2017), we design a probability-aware scoring mechanism $\omega(\widetilde{x}_i)$ that combines prediction uncertainty and prediction accuracy to adaptively compute SDS for each sample $\widetilde{x}_i$:

$$\omega(\widetilde{x}_i) = \omega_1(\widetilde{x}_i) \times \omega_2(\widetilde{x}_i) = (1 + h_\varphi(\widetilde{x}_i))^\lambda \times (1 - \widetilde{g}_i^{gt}), \tag{8}$$

where $\lambda$ is a temperature coefficient, and $\widetilde{g}_i^{gt}$ represents the prediction probability of ground truth. Specifically, SDE assigns higher SDS to hard samples and lower SDS to easy samples, which effectively balances the contribution of each sample to the training objective. By assigning higher scores to hard samples, the model is encouraged to focus more on learning these challenging samples, which can ultimately improve the generalization performance of the FR system. Formally, the SDS weighted classfication loss $\mathcal{L}_{cls}$ can be defined as:

$$\mathcal{L}_{cls} = \omega(\widetilde{x}_i) \times \mathcal{L}_{arc}(\widetilde{x}_i, y_i) \tag{9}$$

During training, to minimize the objective $\mathcal{L}_{cls}$, the model needs to optimize both the SDS $\omega$ and the loss $\mathcal{L}_{arc}$, which brings two benefits: (1) Minimizing $\mathcal{L}_{arc}$ can encourage the model to capture face features with greater generalization ability from diverse training samples. (2) Minimizing SDS $\omega$ can alleviate the damage of hard samples to the latent space's topological structure, which is beneficial to the preservation of topological structure information and the construction of clear decision boundary.

### 4.3 MODEL OPTIMIZATION

To summarize, the overall objective of TopoFR can be formulated as follows:

$$\min_{\mathcal{F}, \mathcal{C}} \mathcal{L}_{cls} + \alpha \mathcal{L}_{sa} \tag{10}$$

where $\alpha$ is hyper-parameter that balances the contributions of the loss $\mathcal{L}_{cls}$ and the loss $\mathcal{L}_{sa}$.

## 5 EXPERIMENTS

### 5.1 DATASETS.

We employ two distinct datasets, namely MS1MV2 Deng et al. (2019) (5.8M facial images, 85K identities) and the recently proposed Glint360K An et al. (2021) dataset (17M facial images, 360K identities), as our training sets. For evaluation, we adopt LFW Huang et al. (2008), AgeDB-30 Moschoglou et al. (2017), CFP-FP Sengupta et al. (2016) and IJB-C Maze et al. (2018) as the benchmarks to test the performance of our models. More training details are placed on **Appendix**.

### 5.2 RESULTS ON MAINSTREAM BENCHMARKS

**Results on LFW, CFP-FP and AgeDB-30.** We employ ResNet-100 He et al. (2016) as the backbone, and adopt MS1MV2 and Glint360K to train our models respectively. To demonstrate the universality of our method, we also provide detailed experimental results of our **TopoFR**[†] model trained by CosFace Wang et al. (2018b). The results are reported in Table 1.

As stated in Refs.Kim et al. (2022); Meng et al. (2021), the performances of existing FR models on these three benchmarks have reached saturation. On MS1MV2 training set, we note that our TopoFR

and TopoFR$^{\dagger}$ models still achieve performance gains. Specially, compared with some cutting-edge competitors (e.g., BroadFace) on these three benchmarks, the proposed TopoFR$^{\dagger}$ trained on MS1MV2 still achieves performance gains on the pose-invariant (0.2% improvement on CFP-FP) and age-invariant (0.04% improvement on AgeDB-30) face verification and becomes a SOTA model.

On Glint360K training set, the proposed TopoFR becomes SOTA model and surpasses ArcFace by 0.04%, 0.39% and 0.41% on three benchmarks, respectively. It is worth mentioning that regardless of the training set used, our TopoFR and TopoFR$^{\dagger}$ models exhibit almost no fluctuation in accuracy across these three benchmarks, which indicates the stability and robustness of our TopoFR.

Table 1: Verification accuracy (%) on LFW, CFP-FP and AgeDB-30 benchmarks. $\dagger$ denotes TopoFR trained by CosFace Wang et al. (2018b).

| Training Data | Method | LFW | CFP-FP | AgeDB-30 |
|---|---|---|---|---|
| MS1MV2 | R100, ArcFace Deng et al. (2019) | 99.77 | 98.27 | 98.15 |
| | R100, CosFace Wang et al. (2018b) | 99.78 | 98.26 | 98.17 |
| | R100, MagFace Meng et al. (2021) | 99.83 | 98.46 | 98.17 |
| | R100, CurricularFace Huang et al. (2020) | 99.80 | 98.37 | 98.32 |
| | R100, BroadFace Kim et al. (2020) | **99.85** | 98.63 | 98.38 |
| | R100, SCF Li et al. (2021b) | 99.82 | 98.40 | 98.30 |
| | R100, AdaFace Kim et al. (2022) | 99.82 | 98.49 | 98.05 |
| | R100, ElasticFace Boutros et al. (2022) | 99.80 | 98.67 | 98.35 |
| | R100, **TopoFR**$^{\dagger}$ | 99.83±0.00 | **98.83**±0.00 | **98.42**±0.00 |
| | R100, **TopoFR** | **99.85**±0.00 | 98.71±0.00 | **98.42**±0.00 |
| Glint360K | R100, ArcFace Deng et al. (2019) | 99.81 | 99.04 | 98.31 |
| | R100, CosFace Wang et al. (2018b) | 99.82 | 99.14 | 98.53 |
| | R100, **TopoFR** | **99.85**±0.00 | **99.43**±0.00 | **98.72**±0.00 |

**Results on IJB-C.** We train our TopoFR on MS1MV2 and Glint360K respectively, and compare with SOTA competitors on IJB-C benchmark, as reported in Table 2. On MS1MV2 training set, our models obtain the best results on "TAR@FAR=1e-4" under different backbones. Specially, our R100 TopoFR model outperforms the advanced method R100 AdaFace Kim et al. (2022) and achieves SOTA performance (96.95%) on "TAR@FAR=1e-4". Furthermore, under the ResNet-50 backbone, our TopoFR and TopoFR$^{\dagger}$ models greatly surpass the previous SOTA method AdaFace by +0.22% and +0.13% on "TAR@FAR=1e-4" respectively, and even beat most ResNet-100 based competitors.

On Glint360K training set, all our models achieve significant performance gains on IJB-C benchmark when compared with the original baselines. For instance, R100 TopoFR outperforms R100 ArcFace by +0.48% and +0.56% on "TAR@FAR=1e-5" and "TAR@FAR=1e-4" respectively. In addition, R200 TopoFR achieves SOTA performance (96.54% and 97.74%) on "TAR@FAR=1e-5" and "TAR@FAR=1e-4" respectively, and becomes a SOTA model. More importantly, the accuracy with a smaller fluctuation range indicates not only the robustness of our models, but also the stability of our strategies under different backbones.

## 5.3 ANALYSIS AND ABLATION STUDY

Due to the limitation of page size, more ablation experiments and analysis are placed on **Appendix**.

**1) Contribution of Each Component:** To investigate the contribution of each component in our model, we employ MS1MV2 as the training set, and compare ArcFace (baseline), and four variants of TopoFR on IJB-C benchmark. The variants of TopoFR are as follows: (1) TopoFR-D, the variant only adds DDA mechanism to ArcFace. (2) TopoFR-A, based on ArcFace, the variant simply aligns the structure of input space and latent space without using DDA. (3) TopoFR-P, the variant fully introduces the PTSA strategy into ArcFace. (4) TopoFR-G, based on TopoFR-P, the variant only uses prediction uncertainty $\omega_1$ modeled by GUM to re-weight each sample. (5) TopoFR-F, based on TopoFR-P, the variant simply applies Focal loss $\omega_2$ to re-weight each sample.

Table 3: Ablation study.

| Training Data | Method | IJB-C(1e-4) |
|---|---|---|
| MS1MV2 | R100, ArcFace | 95.76 |
| | R100, TopoFR-D | 96.12 |
| | R100, TopoFR-A | 96.43 |
| | R100, TopoFR-P | 96.68 |
| | R100, TopoFR-F | 96.73 |
| | R100, TopoFR-G | 96.85 |
| | R100, TopoFR | **96.95** |

The results gathered in Table 3 reflect some observations: (1) Compared with ArcFace, the accuracy of TopoFR-D is improved due to the addition of more training data. (2) TopoFR-A outperforms ArcFace, indicating that simply introducing the loss $\mathcal{L}_{sa}$ can boost model's performance, but it inevitably encounters structure collapse phenomenon. (3) TopoFR-P significantly outperforms

Table 2: Verification accuracy (%) on IJB-C benchmark. † denotes TopoFR trained by CosFace.

| Training Data | Method | IJB-C(1e-6) | IJB-C(1e-5) | IJB-C(1e-4) | IJB-C(1e-3) | IJB-C(1e-2) | IJB-C(1e-1) |
|---|---|---|---|---|---|---|---|
| MS1MV2 | R50, ArcFace Deng et al. (2019) | 80.52 | 88.36 | 92.52 | - | - | - |
| | R50, MagFace Meng et al. (2021) | 81.69 | 88.95 | 93.34 | - | - | - |
| | R50, AdaFace Kim et al. (2022) | - | - | 96.27 | - | - | - |
| | R50, **TopoFR**† | 89.32±0.04 | **94.77**±0.01 | 96.40±0.00 | **97.69**±0.02 | **98.44**±0.00 | **99.12**±0.02 |
| | R50, **TopoFR** | **90.52**±0.02 | 94.71±0.02 | **96.49**±0.00 | 97.62±0.00 | 98.41±0.01 | 99.04±0.00 |
| | R100, CosFace Wang et al. (2018b) | 87.96 | 92.68 | 95.56 | - | - | - |
| | R100, ArcFace Deng et al. (2019) | 85.65 | 92.69 | 95.74 | - | - | - |
| | R100, MV-Softmax Wang et al. (2020) | - | - | 95.20 | - | - | - |
| | R100, CircleLoss Sun et al. (2020) | - | 89.60 | 93.95 | 96.29 | - | - |
| | R100, URL Shi et al. (2020) | - | 95.00 | 96.60 | - | - | - |
| | R100, BroadFace Kim et al. (2020) | 85.96 | 94.59 | 96.38 | - | - | - |
| | R100, CurricularFace Huang et al. (2020) | - | - | 96.10 | - | - | - |
| | R100, MagFace+ Meng et al. (2021) | **90.24** | 94.08 | 95.97 | - | - | - |
| | R100, SCF Li et al. (2021b) | - | 94.78 | 96.22 | - | - | - |
| | R100, DAM-CurricularFace Liu et al. (2021) | | | 96.20 | - | - | - |
| | R100, ElasticFace+ Boutros et al. (2022) | - | - | 96.65 | - | - | - |
| | R100, AdaFace Kim et al. (2022) | - | - | 96.89 | - | - | - |
| | R100, **TopoFR**† | 87.90±0.07 | **95.27**±0.00 | 96.90±0.00 | **97.91**±0.00 | **98.50**±0.00 | **99.20**±0.00 |
| | R100, **TopoFR** | 90.21±0.02 | 95.23±0.00 | **96.95**±0.00 | 97.90±0.00 | 98.49±0.00 | 99.16±0.00 |
| | R200, ArcFace Deng et al. (2019) | 85.75 | 94.67 | 96.53 | 97.60 | 98.25 | 98.90 |
| | R200, **TopoFR**† | 87.90±0.01 | **95.19**±0.00 | **97.12**±0.00 | **98.02**±0.00 | **98.53**±0.00 | **99.06**±0.01 |
| | R200, **TopoFR** | **88.39**±0.01 | 95.15±0.00 | 97.08±0.00 | 97.93±0.00 | 98.47±0.00 | 99.04±0.00 |
| Glint360K | R50, ArcFace Deng et al. (2019) | 88.40 | 95.29 | 96.81 | 97.79 | 98.30 | 99.04 |
| | R50, **TopoFR** | **91.30**±0.02 | **95.99**±0.00 | **97.27**±0.00 | **98.07**±0.00 | **98.71**±0.00 | **99.22**±0.00 |
| | R100, ArcFace Deng et al. (2019) | 86.97 | 96.09 | 97.04 | 98.06 | 98.52 | 99.11 |
| | R100, **TopoFR** | **91.06**±0.01 | **96.57**±0.00 | **97.60**±0.01 | **98.26**±0.00 | **98.76**±0.00 | **99.26**±0.03 |
| | R200, ArcFace Deng et al. (2019) | **89.45** | 95.71 | 97.20 | 97.98 | 98.38 | 99.09 |
| | R200, **TopoFR** | 86.84±0.03 | **96.54**±0.00 | **97.74**±0.00 | **98.31**±0.01 | **98.80**±0.00 | **99.20**±0.01 |

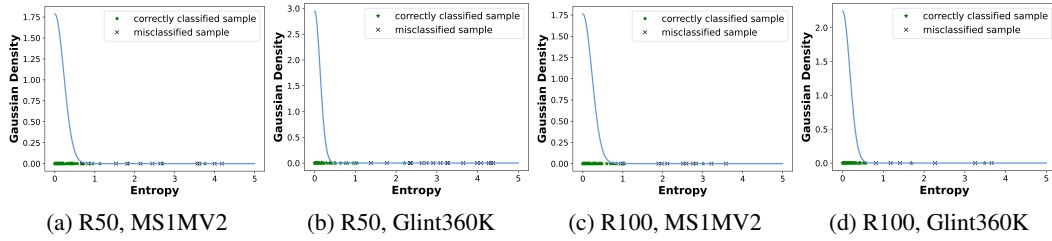

(a) R50, MS1MV2  (b) R50, Glint360K  (c) R100, MS1MV2  (d) R100, Glint360K

Figure 4: The estimated Gaussian density (blue curve) w.r.t the entropy of classification prediction. Green marker ⋆ and black marker × represent the entropy of correctly classified sample and misclassified sample, respectively.

TopoFR-D and TopoFR-A, indicating that preserving the topological structure information using our PTSA strategy can greatly improve FR model's generalization. (4) TopoFR outperforms TopoFR-F and TopoFR-G, which not only demonstrates the effectiveness of our SDE strategy, but also indicates that the prediction uncertainty $\omega_1$ is complementary to the Focal loss $\omega_2$ in mining hard samples.

**2) Effectiveness of GUM:** To visually demonstrate the effectiveness of GUM in mining hard samples, we present the estimated Gaussian density of the prediction entropy during training in Figure 4. These curves show that the entropy of misclassified face samples (represented by black crosses) usually have rather low Gaussian density (i.e., high posterior probability $h_\varphi$), thus can be easily detected. Note that even if some misclassified samples have small entropy (i.e., high Gaussian density and low posterior probability $h_\varphi$), their SDS $\omega$ can still be corrected by the Focal loss $\omega_2$.

**3) Generalization Performance of PTSA:** To demonstrate the superior generalization ability of our PTSA strategy in preserving structure information, we investigate the topological structure discrepancy between the input space and the latent space of TopoFR and its variant TopoFR-A on IJB-C benchmark. Note that TopoFR-A directly utilizes PH to align the topological structures of two spaces. The results illustrated in Figure 5 indicate that: 1) Directly using PH to align the topological structures of two spaces does not effectively reduce the structure discrepancy, as the model suffer from the structure collapse phenomenon; 2) PTSA strategy can effectively align the topological structures of two spaces and address this structure collapse phenomenon. Remarkably, Figures 5c and 5d show that our TopoFR models trained on Glint360K dataset almost perfectly preserve struc-

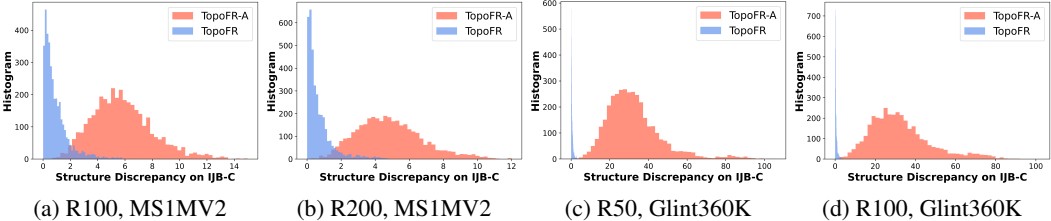

| (a) R100, MS1MV2 | (b) R200, MS1MV2 | (c) R50, Glint360K | (d) R100, Glint360K |

Figure 5: The topological structure discrepancy of TopoFR and variant TopoFR-A under different backbones and training datasets (i.e., [Backbone, Training dataset]). Note that variant TopoFR-A directly adopts PH to align the topological structures of two spaces. It is worth mentioning that our TopoFR models trained with Glint360K dataset almost perfectly align the topological structures of the input space and the latent space on the IJB-C benchmark (i.e., the blue histogram almost converges to a straight line).

ture information of the input space in their latent features, thereby verifying the generalization ability of PTSA strategy.

## 6 CONCLUSION

This paper proposes a novel FR framework called TopoFR that aims to encode the critical structure information hidden in large-scale face dataset into the latent space. Specially, TopoFR leverages two key strategies: structure alignment strategy PTSA and hard sample mining strategy SDE. PTSA employs persistent homology to reduce the topological structure discrepancy between the input space and latent spaces, effectively mitigate structure collapse phenomenon and preserving structure information. SDE accurately identifies hard samples by adaptively computing SDS for each sample, and guides the model to prioritize optimizing these samples, mitigating their adverse impact on the latent space structure. Comprehensive experiments and analysis demonstrate the superiority of our proposed TopoFR on various popular face benchmarks.

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

## A APPENDIX

### A.1 IMPLEMENTATION DETAILS

Our models are trained using Pytorch on 4 NVIDIA Tesla A100 GPUs, and a mini-batch of 128 images is assigned for each GPU. We crop all images to 112×112, following the same setting as in ArcFace Deng et al. (2019). For the backbone, we adopt ResNet50, ResNet100 and ResNet200 He et al. (2016) as modified in Deng et al. (2019). We follow Deng et al. (2019) to employ ArcFace ($s = 64$ and $m = 0.5$) as the basic classification loss to train the TopoFR model. For the TopoFR$^\dagger$ model trained by CosFace Wang et al. (2018b), we set the scale $s$ to 64 and the cosine margin $m$ of CosFace to 0.4. To optimize the models, we use Stochastic Gradient Descent (SGD) optimizer with momentum of 0.9 for both datasets. The weight decay for MS1MV2 is set to 5e-4 and 1e-4 for Glint360K. The initial learning rate is set to 0.1 for both datesets. In terms of the balance coefficient $\alpha$, we choose $\alpha = 0.1$ for experiments on R50 TopoFR, and $\alpha = 0.05$ for experiments on R100 TopoFR and R200 TopoFR. During training, we apply DDA mechanism with a certain probability. Specially, for an original input sample $x$, the probability of it undergoing DDA is $\xi$, and the probability of it remaining unchanged is $1 - \xi$. For the hyper-parameter $\xi$, we choose $\xi = 0.2$. We utilize the Ripser package Tralie et al. (2018) to compute persistence diagrams. Notably, in our method, we focus on preserving the 0-dimension homology $H_0$ in the topological structure alignment loss $\mathcal{L}_{sa}$. Because preliminary experiments demonstrated that using the 1-dimension or higher-dimension homology only increases model's training time without clear performance gains.

### A.2 MORE ABLATION EXPERIMENTS AND ANALYSIS

**1) Effectiveness of DDA:** To demonstrate the effectiveness of DDA in expanding the structure diversity of the latent space, we employ MS1MV2 as the training set, and compare TopoFR-P with its multiple variants on the IJB-C benchmark. The variants of TopoFR-P are as follows: (1) TopoFR-P ($\mathcal{A}_{i'}$), the variant denotes that only one data augmentation operation $\mathcal{A}_{i'}$ is included in the list $\mathcal{A}$. (2) TopoFR-P ($\mathcal{A}_{i'} + \mathcal{A}_j$), the variant indicates that two data augmentation operations $\mathcal{A}_{i'}$ and $\mathcal{A}_j$ are included in the list $\mathcal{A}$. (3) TopoFR-P ($\mathcal{A}_{i'} + \mathcal{A}_j + \mathcal{A}_{j'}$), the variant denotes that three data augmentation operations $\mathcal{A}_{i'}$, $\mathcal{A}_j$ and $\mathcal{A}_{j'}$ are included in the list $\mathcal{A}$.

The results gathered in Table 4 demonstrate that using a variety of data augmentation strategies can more effectively perturb the topological structure of the latent space and mitigate the structure collapse phenomenon, leading to a significant improvement in the generalization performance of FR model.

Table 4: Effectiveness of DDA. TAR@FAR=1e-4 is reported on the IJB-C benchmark.

| Training Data | Method | IJB-C(1e-4) |
|---|---|---|
| MS1MV2 | R100, TopoFR-P ($\mathcal{A}_1$) | 96.48 |
| | R100, TopoFR-P ($\mathcal{A}_2$) | 96.36 |
| | R100, TopoFR-P ($\mathcal{A}_3$) | 96.40 |
| | R100, TopoFR-P ($\mathcal{A}_4$) | 96.42 |
| | R100, TopoFR-P ($\mathcal{A}_1 + \mathcal{A}_2$) | 96.58 |
| | R100, TopoFR-P ($\mathcal{A}_1 + \mathcal{A}_3$) | 96.50 |
| | R100, TopoFR-P ($\mathcal{A}_1 + \mathcal{A}_4$) | 96.56 |
| | R100, TopoFR-P ($\mathcal{A}_2 + \mathcal{A}_3$) | 96.45 |
| | R100, TopoFR-P ($\mathcal{A}_2 + \mathcal{A}_4$) | 96.41 |
| | R100, TopoFR-P ($\mathcal{A}_3 + \mathcal{A}_4$) | 96.46 |
| | R100, TopoFR-P ($\mathcal{A}_1 + \mathcal{A}_2 + \mathcal{A}_3$) | 96.63 |
| | R100, TopoFR-P ($\mathcal{A}_1 + \mathcal{A}_2 + \mathcal{A}_4$) | 96.59 |
| | R100, TopoFR-P ($\mathcal{A}_1 + \mathcal{A}_3 + \mathcal{A}_4$) | 96.61 |
| | R100, TopoFR-P ($\mathcal{A}_2 + \mathcal{A}_3 + \mathcal{A}_4$) | 96.57 |
| | R100, TopoFR-P | **96.68** |

**2) Comparison with Previous Hard Sample Mining Strategies:** To further demonstrate the superiority of our SDE strategy in mining hard samples, we compare it with existing hard sample mining strategies, including MV-Softmax Wang et al. (2020) and CurricularFace Huang et al. (2020). The

results presented in Table 5 indicate that our SDE strategy is better able to measure sample difficulty and facilitate the alignment of topological structures between the original input space and the perturbed latent space. This is because the SDE comprehensively considers the prediction uncertainty and prediction probability when mining hard samples.

Table 5: Comparison with Previous Hard Sample Mining Strategies.

| Training Data | Method | IJB-C (1e-4) |
|---|---|---|
| MS1MV2 | R100, TopoFR-P | 96.68 |
| | R100, TopoFR-P + MV-Softmax | 96.74 |
| | R100, TopoFR-P + CurricularFace | 96.79 |
| | R100, TopoFR | **96.95** |

**3) Effectiveness of ISA:** Previous works often use Bottleneck distance and p-Wasserstein distance to measure the distance between persistence diagrams $\mathcal{D}^{\mathcal{X}}$ and $\mathcal{D}^{\widetilde{\mathcal{Z}}}$ Mileyko et al. (2011); Turner et al. (2014). Concretely, the Bottleneck distance is defined as $\mathcal{L}_{\infty}(\mathcal{D}^{\mathcal{X}}, \mathcal{D}^{\widetilde{\mathcal{Z}}}) = \inf_{\kappa:\mathcal{D}^{\mathcal{X}} \to \mathcal{D}^{\tilde{z}}} \sup_{\varpi \in \mathcal{D}^{\mathcal{X}}} \|\varpi - \kappa(\varpi)\|_{\infty}$, with $\kappa$ ranging over all bijections between sets of persistent intervals in diagrams $\mathcal{D}^{\mathcal{X}}$ and $\mathcal{D}^{\widetilde{\mathcal{Z}}}$, and $\|\cdot\|_{\infty}$ denotes the $\infty-$norm. Equivalently, the p-Wasserstein distance is defined as $\mathcal{L}_p(\mathcal{D}^{\mathcal{X}}, \mathcal{D}^{\widetilde{\mathcal{Z}}}) = (\inf_{\kappa:\mathcal{D}^{\mathcal{X}} \to \mathcal{D}^{\tilde{z}}} \sum_{\varpi \in \mathcal{D}^{\mathcal{X}}} \|\varpi - \kappa(\varpi)\|_{\infty}^{p})^{1/p}$. Furthermore, some studies Cuturi (2013); Carriere et al. (2017); Kerber et al. (2017); Chen & Wang (2021) have also attempted to accelerate the computation of p-Wasserstein distances between persistence diagrams. However, the Bottleneck distance metric and p-Wasserstein distance metric are sensitive to outliers Horak et al. (2021); Bertsekas (1981); Som et al. (2018) and will increase the training time of the FR models. This makes them unsuitable for FR tasks with extremely large-scale datasets.

To demonstrate the stability and computational efficiency of our ISA strategy, we employ MS1MV2 as the training set, and compare TopoFR and its two variants on the IJB-C benchmark. The variants of TopoFR are as follows: (1) TopoFR-B, the variant uses Bottleneck distance to compute the discrepancy between persistence diagrams. (2) TopoFR-W, the variant adopts 1-Wasserstein distance to measure the discrepancy between persistence diagrams.

We provide the average training time per epoch of these models and their recognition performance on "TAR@FAR=1e-4" of the IJB-C benchmarks. The results presented in Table 6 reflect the following observations: (1) TopoFR outperforms TopoFR-B and TopoFR-W on "TAR@FAR=1e-4", indicating the robustness of our ISA to outliers. (2) Compared with TopoFR-B and TopoFR-W, our proposed TopoFR has a shorter training time, demonstrating the computational efficiency of ISA.

Table 6: Comparison of TopoFR using Different Distance Metrics.

| Training Data | Method | Average Training Time / Epoch | IJB-C (1e-4) |
|---|---|---|---|
| MS1MV2 | R100, TopoFR-B | 4312.56s | 96.76 |
| | R100, TopoFR-W | 4127.29s | 96.81 |
| | R100, TopoFR | **2729.28s** | **96.95** |

**4) Parameter Sensitivity:** To demonstrate the effect of the hyper-parameters $\alpha$ (i.e., the balance coefficient of the ISA loss $\mathcal{L}_{sa}$) and $\xi$ (i.e., the probability of DDA mechanism), we conduct additional experiments by setting different values of $\alpha$ and $\xi$, respectively. We use MS1MV2 dataset to train the R50 TopoFR and R100 TopoFR models, and evaluate their performance on the IJB-C benchmark. The results on "TAR@FAR=1e-4" are shown in Figures 6a and 6b. We can observe that as $\alpha$ increases, the accuracy first rises and then falls. This is because an overly large $\alpha$ will cause the model to focus more on the structure alignment and less on the classification learning of the samples during training. Additionally, setting the parameter $\xi$ to a high value may significantly disturb the topological structure of the latent space, which can make it difficult to achieve convergence of the ISA loss $\mathcal{L}_{sa}$.

**5) Visualization of Hard Samples:** We conduct a visual comparison experiment to visualize some hard samples that can be correctly classified by our method but cannot be correctly classified by existing method. The visualization results illustrated in Figure 7 reflect the following observations: (1) Hard samples are usually blurry, low-contrast, occluded, or in unusual poses, so they are easily misclassified by existing method such as ArcFace model. (2) The ArcFace model assigns equal weight (i.e., 1) to each sample. While our TopoFR model utilizes SDE strategy to adaptively assign

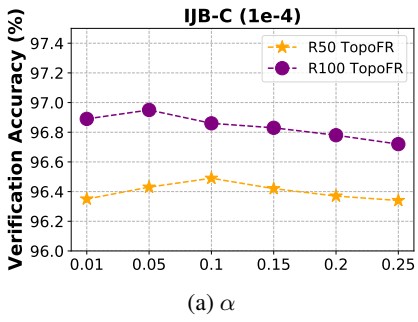
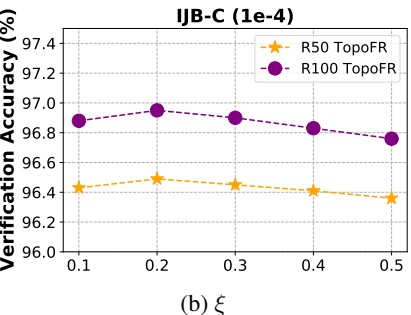

Figure 6: Parameter sensitivity analysis. (a) The effect of the hyper-parameter $\alpha$. (b) The effect of the hyper-parameter $\xi$.

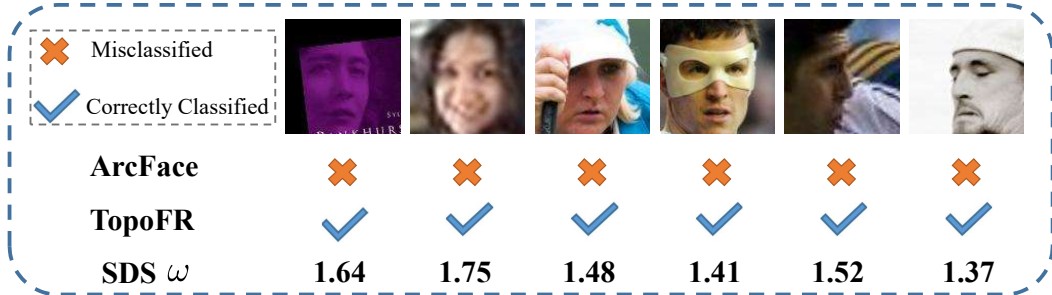

Figure 7: Visualization of hard samples.

weight (i.e., SDS $\omega$) to each sample based on its prediction uncertainty $\omega_1$ and prediction accuracy $\omega_2$. Specially, SDE will assign higher SDS $\omega$ to hard samples, which can encourage the model to extract robust face features from these challenging samples, thereby effectively improving the model's generalization performance.

**5) Training Time:** For detailed training time analysis, please refer to the Table 7. Due to the introduction of the structure alignment strategy PTSA and hard sample mining strategy SDE, our TopoFR models require longer training time (1.16x). Specially, compared to the vanilla R50 ArcFace model, our R50 TopoFR model requires about 2 seconds more training time per 100 steps, which does not significantly increase the training time but brings a large performance gain. And our R100 topoFR model requires about 3 seconds extra training time per 100 steps than the vanilla R100 ArcFace model, which is not a major increase in traing time but leads to a significant performance advantage. While for inference computation head, our method performs consistently with that of vanilla ArcFace model, since we adopt the same network architecture and data pre-process module.

Moreover, the results in Table 7 indicate that introducing SDE strategy (i.e., GUM) leads to significant performance improvements with only a small increase in training time (i.e., R50 backbone: 0.2s / 100 steps, R100 backbone: 0.25s / 100 steps), which is reasonable. Therefore, the addition of GUM does not bring too much computational burden and does not significantly increase the training time.

Table 7: Detailed training time of models.

| Training Data | Method | Average Training Time / 100 steps | Average Training Time / Epoch | IJB-C (1e-4) |
|---|---|---|---|---|
| MS1MV2 | R50, ArcFace | 15.33s | 1743.32s | 92.52 |
|  | R50, ArcFace + PTSA | 17.57s | 1999.19s | 96.25 |
|  | R50, ArcFace + PTSA + SDE (**R50, TopoFR**) | 17.77s | 2020.80s | **96.49** |
| MS1MV2 | R100, ArcFace | 20.16s | 2292.59s | 95.74 |
|  | R100, ArcFace + PTSA | 23.10s | 2626.93s | 96.68 |
|  | R100, ArcFace + PTSA + SDE (**R100, TopoFR**) | 23.35s | 2655.36s | **96.95** |

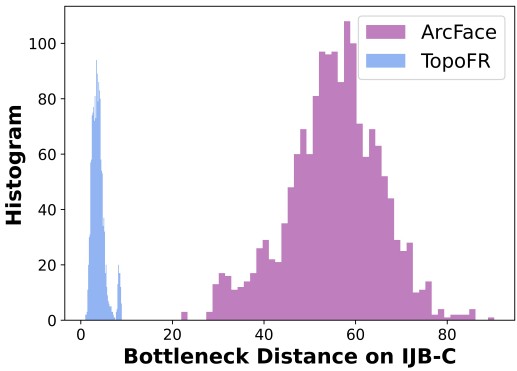

Figure 8: The topological structure discrepancy (i.e., measured by the **Bottleneck distance metric**) of R50 TopoFR and R50 ArcFace on the IJB-C benchmark.

**6) Comparison of Structure Discrepancy:** To further demonstrate the effectiveness of our PTSA strategy in preserving structure information, we utilize the **Bottleneck distance metric** Turner et al. (2014); Mileyko et al. (2011) to investigate the topological structure discrepancy between the input space and the latent space of ArcFace and TopoFR on IJB-C benchmark, as illustrated in Figure 8. We can observe that our TopoFR model significantly reduces the structure discrepancy (i.e., measured by the Bottleneck distance metric) between two spaces compared to the vanilla ArcFace model, and effectively preserves the structure information hidden in the large-sclae dataset.

