# OpenReview forum: "TopoFR: A Closer Look at Topology Alignment on Face Recognition"
_ICLR.cc/2024/Conference — Submitted to ICLR 2024_

### Official Review · Reviewer_21zk · 2023-10-29

**Soundness:** 3 good
**Presentation:** 3 good
**Contribution:** 3 good
**Rating:** 6
**Confidence:** 4

**Summary:**

In this paper, the authors propose a topological structure alignment method for face recognition. Benefiting from the invariant topology structure in latent space, the authors propose diverse data augmentation (DDA) and invariant structure alignment (ISA) to optimize face embeddings. Experiments on various face recognition datasets achieve state-of-the-art performance and the code is provided

**Strengths:**

1. The authors propose a topological structure alignment method for face recognition.
2. The authors propose diverse data augmentation (DDA) and invariant structure alignment (ISA) to optimize face embeddings.
3. Experiments on various face recognition datasets achieve state-of-the-art performance.
4. Code is provided.

**Weaknesses:**

1. For diverse data augmentation and invariant structure alignment, the ISA loss only uses original face images and the corresponding augmented images for optimization. It is similar with discriminative self-supervised learning. However, for face recognition, the intra-class relations are also important.  Could the authors give some detailed comparisons and analyses to optimize the topological structure alignment with intra-class and inter-class distances?

2. Since face recognition has made great progress in recent years, the generalized performance is important. I recommend the authors to pay more attentions on the current SOTA on FRVT (https://pages.nist.gov/frvt/html/frvt11.html) and MFR-Ongoing (http://iccv21-mfr.com/#/leaderboard/academic). It is not difficult to obtain a comparable performance on FRVT and MFR-Ongoing by only using the WebFace260M dataset. For face recognition community, it is more meaningful to certificate that the proposed method can improve the perforamnce of generalized evalutions on age/pose/cross-domain face recognition.

**Questions:**

See weaknesses.

---

> ### Author Response · Authors · 2023-11-22
> **Response to Reviewer 21zk**
>
> **Q1: For diverse data augmentation and invariant structure alignment, the ISA loss only uses original face images and the corresponding augmented images for optimization. It is similar with discriminative self-supervised learning. However, for face recognition, the intra-class relations are also important. Could the authors give some detailed comparisons and analyses to optimize the topological structure alignment with intra-class and inter-class distances?**
>
> **A1:** Based on your advice, we conducted two pairs of ablative experiments to validate the relationship between topological structure alignment and intra-class distance constraint as well as inter-class distance constraint. The results are gathered in the following Table B1.
>
> We can find that integrating topological structure alignment with single inter-class or intra-class distance constraint can both obtain additional significant performance gains. This indicates that topological structure alignment can provide the extra structure information of intra-class and inter-class relationships, thereby establishing clearer decision boundaries.
>
> **Table B1: Verification accuracy (\%) on IJB-C.**
> | Training Data | Method | IJB-C(1e-4) |
> | ------ | ------ | ------ |
> |  | R100 ArcFace (w/o intra-class) | 94.16 |
> | MS1MV2 | R100 ArcFace (w/o intra-class) + topological constraint | **94.86** |
> |  | R100 ArcFace (w/o inter-class) | 94.47 |
> |  | R100 ArcFace (w/o inter-class) + topological constraint | **95.12** |
>
> **Q2: Since face recognition has made great progress in recent years, the generalized performance is important. I recommend the authors to pay more attentions on the current SOTA on FRVT and MFR-Ongoing. It is not difficult to obtain a comparable performance on FRVT and MFR-Ongoing by only using the WebFace260M dataset. For face recognition community, it is more meaningful to certificate that the proposed method can improve the perforamnce of generalized evalutions on age/pose/cross-domain face recognition.**
>
> **A2:** Thanks for your valuable advice.
>
> We utilize the WebFace260M dataset as our training set and train our proposed TopoFR model using 64 V100 GPUs. The entire process takes us a significant amount of time. In order to improve the model's generalization performance on MFR-Ongoing, we adopt the mask augmentation operation. Based on your suggestions, we have already submitted our TopoFR model to the MFR-Ongoing. **We can observe that our TopoFR model ranks second on MFR-Ongoing, surpassing the recently proposed method UniTSFace [1]**.
>
> For the FRVT, we will submit the model within the next two weeks due to its cumbersome submission process, but based on past experience, it takes a considerable amount of time (approximately 3 months to 6 months) to obtain results and be listed on the leaderboard.
>
> In conclusion, we would like to express our heartfelt appreciation for your recognition of our method and the valuable suggestions you have provided.
>
> Ref.[1] UniTSFace: Unified Threshold Integrated Sample-to-Sample Loss for Face Recognition. Neurips 2023.

---

### Official Review · Reviewer_7Xi6 · 2023-10-30

**Soundness:** 3 good
**Presentation:** 3 good
**Contribution:** 3 good
**Rating:** 6
**Confidence:** 3

**Summary:**

The paper introduces the combination of a topological loss directly adopted from [Moor20] (Topological Autoencoders, ICML 2020) and a novel hard examples mining strategy to the problem of face recognition within the margin-based softmax loss approach.  The paper’s main claim is that the topology preservation of the data’s representation learned by a neural network would improve the model is supported with experiments on several benchmark datasets.
In particular, the authors introduce three observations they claim to be novel, supporting them with empirical evidence only, with the illustrations of the results of computational experiments:

a) data’s topological complexity increases with its amount, illustrated by the persistence diagrams,

b) persistent homology-based distance between input data and its representation learned by a neural network increases with data’s amount, illustrated by the distance histograms evaluated for batches of sizes 256, 512 and 1024,

c) persistent homology-based distance between input data and its representation learned by a neural network decreases with the network’s depth, illustrated by the distance histograms evaluated for batch of size 128 for ResNets of three several depths.

An extensive ablation study shows the contribution of each component with the topology-based loss improves on 0.95-0.69 percentage points (with and without data’s augmentation), while adding other components brings only 0.27 percentage points improvement.

**Strengths:**

1. Interesting implementation of topological data analysis and persistence diagrams in face recognition
2. Strong experimental results that demonstrate benefits over existing methods

**Weaknesses:**

1.	The observation (a) is not novel, as it was theoretically investigated before [Kahle11,Bobrowski18], more on this later. The purpose of providing this observation is not completely understood. Is the author’s point that real world data, face images in particular, have complex topology, that need to be preserved? Then illustrating it with persistence diagrams just having more homology classes representatives as the amount of data’s increase is not enough. One would observe that with any random data of size n in dimension d, as it was shown that the expected k-th Betti number is $\mathbb{E}[\beta_k^{VR}(r)] = c_k n(nr^d)^{2k+1}$, with only the constant $c_k$ actually depends on data’s distribution [Kahle11]. In other words, the more data you have the more configurations of r-thickened points could form k-cycles. So what matters is the distribution of persistence diagrams as they would be different for random and real-world data.
2.	The observation b) could be done by not properly normalizing the distance. BTW, it is better to use the same colors for batches of the same size for an improved comprehension, and to use the same size of the batch for illustrations of c) and one of the batch sizes of b).
3.	The claim that optimal transport-based distances have high time complexity to be used with real-world data is too loud, with the Wasserstein distance is only 1.5 times slower as shown in Table 6.
4.	The significance of the results (Table 1,2) is not analyzed. It would be better to report standard deviation instead of the mean accuracy only
5.	Minor comments:
5.1.	It would be better to show the author’s attempt to train ArcFace by their own instead of using the pre-trained model in ablation study (Table 3). It is possible that they get higher accuracy, and the difference with the proposed technique will be lower.
5.2.	the software used to compute persistence diagrams is not stated in the main text,
5.3.	the homology dimension(s) H_* to be preserved by the topological loss are not stated in the main text;
5.4.	for the optimal transport-based distances only the time complexity O(n^3) is stated, yet the approximations in O(n^2) [Cuturi13], O(n log n) [Carriere17], O(n^~1.6) — empirically estimated [Kerber17], and near linear [Chen21] time exists, with at least first two are differentiable.

[Kahle11] Random Geometric Complexes, Discrete & Computational Geometry (2011)
[Bobrowski18] Topology of Random Geometric Complexes: A Survey, J. Appl. and Comput. Topology (2018)
[Cuturi13] Sinkhorn Distances: Lightspeed Computation of Optimal Transport (2013)
[Carriere17] Sliced Wasserstein Kernel for Persistence Diagrams, ICML (2017)
[Kerber17] Geometry Helps to Compare Persistence Diagrams, ACM Journal of Experimental Algorithmics (2017)
[Chen21] Approximation Algorithms for 1-Wasserstein Distance between Persistence Diagrams, International Symposium on Experimental Algorithms (2021)

**Questions:**

1. What results of the proposed method in Table 1,2 are significantly better than the current state-of-the-art methods?
2. The authors mainly report results for R100 backbone, but Table 7 contains results for R50 only. hHat is the difference in training time with vanilla ArcFace for R100?

---

> ### Author Response · Authors · 2023-11-22
> **Response to Reviewer 7Xi6 Q1-Q2**
>
> **Q1: The observation (a) is not novel, as it was theoretically investigated before [Kahle11,Bobrowski18], more on this later. The purpose of providing this observation is not completely understood. Is the author’s point that real world data, face images in particular, have complex topology, that need to be preserved? Then illustrating it with persistence diagrams just having more homology classes representatives as the amount of data’s increase is not enough. One would observe that with any random data of size n in dimension d, as it was shown that the expected k-th Betti number is $\mathbb{E}[\beta_{k}^{VR}(r)]=c_{k}n(nr^{d})^{2k+1}$, with only the constant $c_{k}$
> actually depends on data’s distribution [Kahle11]. In other words, the more data you have the more configurations of r-thickened points could form k-cycles. So what matters is the distribution of persistence diagrams as they would be different for random and real-world data.**
>
> **A1:** Thanks for your valuable advice. The purpose of presenting observation (a) is to demonstrate that the structure information of the latent space in FR models has been severely destroyed, and previous works on FR have overlooked this issue.
> We are the first to explore the topological structure alignment in FR task, and experimental results also demonstrate the SOTA performance of our method.
> Furthermore, as we stated, as the amount of data increases, the topological structure discrepancy becomes increasingly larger. FR models are typically trained on large-scale datasets (i,e., MS1MV2: 5.8M face images; Glint360k: 17M face images), so preserving the structure information of these data is very meaningful, which can effectively enhance the model's generalization performance.
>
> In addition, as you said " the expected k-th Betti number is $\mathbb{E}[\beta_{k}^{VR}(r)]=c_{k}n(nr^{d})^{2k+1}$" and "the more data you have the more configurations of r-thickened points could form k-cycles", they are consistent with our observation (a). It is worth mentioning that these topological works [1,2] only remain at the theoretical level and conduct experiments and analysis solely on random points, while we apply them to the real-world computer vision task (i.e., FR task) with large-scale datasets and achieve notable performance gains.
>
> We'd like to once again thank you very much for informing us about this theory. We have added the citation of this theory in the revised version to better support our motivation and observation.
> We will also express our gratitude in the acknowledgment section of the final version of the paper for the theoretical support you have provided to our work.
>
> Ref.[1] Random geometric complexes. DCG 2011.
>
> Ref.[2] Topology of random geometric complexes: a survey. JACT 2018.
>
> **Q2: The observation b) could be done by not properly normalizing the distance. BTW, it is better to use the same colors for batches of the same size for an improved comprehension, and to use the same size of the batch for illustrations of c) and one of the batch sizes of b).**
>
> **A2:** Thanks for your valuable advice. In order to more intuitively visualize the structure discrepancy between two spaces before and after the structure alignment (as shown in Figures 2(b), 2(d) and 5), we did not normalize the distance.
> Notably, even after normalizing the distance, the absolute structure discrepancy may become smaller, but the relative structure discrepancy still persist.
>
> Moreover, thank you very much for reminding us about this color issue.
> The results reported for Figures 2 (b), (c), and (d) are obtained using the same batch size (i.e., 128). Following your advice, we have made adjustments to Figure 2 and its caption section to better assist readers in understanding them.
> Please refer to the revised version. The modified sections are highlighted in blue.

---

> ### Author Response · Authors · 2023-11-22
> **Response to Reviewer 7Xi6 Q3-Q5**
>
> **Q3: The claim that optimal transport-based distances have high time complexity to be used with real-world data is too loud.**
>
> **A3:** Thanks for reminding us this issue. Indeed, as shown in Table 6, the 1-Wasserstein distance computation takes approximately 1.5 times slower than our ISA strategy during each epoch.
> In other words, the computation of the 1-Wasserstein distance takes an additional 1398.01 seconds compared to our ISA strategy in each iteration. It is worth mentioning that, in the field of FR, a reliable FR model typically requires training for more than 30 or even more than 35 epochs, resulting in significant time accumulation. Therefore, in this case, using 1-Wasserstein distance will significantly increase the training time (i.e., $1398.01 \times 30 = 41940.3$ seconds).
>
> Furthermore, it is worth mentioning that if we adopt an extremely large-sclae face dataset named WebFace260M [3] as training set, training an ArcFace baseline model would takes approximately 10 days using 32 V100 GPUs. In this scenario, using 1-Wasserstein distance would significantly increase the training time.
>
> According to your advice, we revise the viewpoint as follows: "These two distance metrics are sensitive to outliers and will increase the training time of the FR model."
>
> Ref.[3] Webface260M: A benchmark for million-scale deep face recognition. TPAMI 2022.
>
> **Q4: The significance of the results (Table 1,2) is not analyzed. It would be better to report standard deviation instead of the mean accuracy only.**
>
> **A4:** Because there are no standard deviation in the original papers using these competitive methods, we eliminated the standard deviation of our experimental results in the original manuscript to unify the data format.
> For the TopoFR results in Tables 1 and 2, we repeat the experiments 3 times with various seeds and only report the average accuracy. According to your valuable advice, we have included the standard deviation of our proposed method in Tables 1 and 2 of the revised paper.
> Furthermore, we have added corresponding analysis. Please refer to the revised paper.
>
> **Q5: Minor comments: 5.1. It would be better to show the author’s attempt to train ArcFace by their own instead of using the pre-trained model in ablation study (Table 3). 5.2. the software used to compute persistence diagrams is not stated in the main text, 5.3. the homology dimension(s) $H_*$ to be preserved by the topological loss are not stated in the main text;
> 5.4. for the optimal transport-based distances only the time complexity $O(n^3)$ is stated, yet the approximations in $O(n^2)$ [Cuturi13], $O(n log n)$ [Carriere17], $O(n^{\sim1.6})$ empirically estimated [Kerber17], and near linear [Chen21] time exists, with at least first two are differentiable.**
>
> **A5:** Thanks for your valuable suggestion.
>
> (1) **About ablation study:** Base on your suggestion, we retrain the R100 ArcFace model three times with different random seeds and report the average accuracy on "TAR@FAR=1e-4" (i.e., 95.76\%).
> We can find that the newly obtained accuracy (i.e., 95.76\%) is almost the same as that of the pre-trained model (i.e., 95.74\%).
> It is worth noting that in the original ArcFace paper [4], the reported verification accuracy on "TAR@FAR=1e-4" is 95.60\%.
> We obtain the accuracy of 95.74\% from Ref.[5], which is the result re-implemented by the authors themselves.
> Therefore, the results of the ablation study in Table 3 are sufficient to demonstrate the effectiveness of our methods.
> Please refer to the Table 3 in the revised paper.
>
> (2) **About software:** We utilize the Ripser package [6] to compute persistence diagrams. We have included this detail in Section A.1.
>
> (3) **About the homology dimension:** Thanks for reminding us of this issue. In our method, we focus on preserving the 0-dimension homology $H_{0}$ in the topological structure alignment loss. Because preliminary experiments demonstrated that using the 1-dimension or higher-dimension homology only increases model's training time without clear performance gains.
> We have included this detail in Section A.1.
>
> (4) **About the time complexity of optimal transport:** Thanks for your value advice.
>
> We have stated and cited the advancements in optimal transport-based distances in the appendix of our paper. However, due to the difficulty of reproducing these algorithms and the fact that most existing persistent homology packages (including the Ripser package we utilized) only implement the classic optimal transport with a complexity of $\mathcal{O}(n^{3})$, we will only compare our strategy with classical optimal transport and refrain from further discussion on this matter.
>
> Ref.[4] Arcface: Additive angular margin loss for deep face recognition. CVPR 2019.
>
> Ref.[5] Magface: A universal representation for face recognition and quality assessment. CVPR 2021.
>
> Ref.[6] Ripser.py: A lean persistent homology library for python. JOSS 2018.

---

> ### Author Response · Authors · 2023-11-22
> **Response to Reviewer 7Xi6 Q6-Q7**
>
> **Q6: What results of the proposed method in Table 1,2 are significantly better than the current state-of-the-art methods?**
>
> **A6:** Thanks for your valuable advice.
>
> Table 1 reports the models' accuracy on LFW, CFP-FP, and AgeDB-30 benchmarks. Note that LFW, CFP-FP, and AgeDB-30 are small face verification datasets. As stated in Refs.[7,8,9], the performances of existing FR models on these three datasets have reached saturation, making the gain less pronounced.
> On MS1MV2 training set, our TopoFR and TopoFR$^{\dagger}$ models outperform previous SOTA methods BroadFace [10] and ElasticFace [11], and achieve the best performance on these three benchmarks.
> Moreover, compared with competitors on these three benchmarks, the proposed TopoFR trained on Glint360K still achieves performance gains on the pose-invariant (0.3\% improvement on CFP-FP) and age-invariant (0.2\% improvement on AgeDB-30) face verification and becomes a SOTA model.
>
> Furthermore, in the field of FR, IJB-C is currently the most general benchmark dataset for evaluating the generalization performance of FR models. And "TAR@FAR=1e-4" is the most critical evaluation metric.
> As shown in Table 2 of the paper, all our models achieve significant performance gains on IJB-C benchmarks across all metrics. In particular, our proposed TopoFR and TopoFR$^{\dagger}$ models greatly outperform the SOTA method AdaFace[7] and achieve the best performance on "TAR@FAR=1e-4", this can demonstrate the superiority and robustness of our proposed method.
>
> Note that the Glint360K is a recently proposed extremely large-scale dataset on which few studies currently conduct experiments. Therefore, we only re-implement methods AcrFace and CosFace on it.
>
> Based on your valuable suggestions, we have conducted a more detailed analysis of the SOTA performance of our models. Please refer to the revised paper.
>
> In addtion, following Reviewer 21zk's advice, we have also submitted our TopoFR model to the MFR-Ongoing platform (http://iccv21-mfr.com/#/leaderboard/academic). We can observe that our TopoFR model ranks second on MFR-Ongoing, surpassing the recently proposed method UniTSFace [12].
>
> Ref.[7] Adaface: Quality adaptive margin for face recognition. CVPR 2022.
>
> Ref.[8] Magface: A universal representation for face recognition and quality assessment. CVPR 2021.
>
> Ref.[9] Curricularface: adaptive curriculum learning loss for deep face recognition. CVPR 2020.
>
> Ref.[10] Broadface: Looking at tens of thousands of people at once for face recognition. ECCV 2020.
>
> Ref.[11] Elasticface: Elastic margin loss for deep face recognition. CVPR 2022.
>
> Ref.[12] UniTSFace: Unified Threshold Integrated Sample-to-Sample Loss for Face Recognition. Neurips 2023.
>
> **Q7: The authors mainly report results for R100 backbone, but Table 7 contains results for R50 only. What is the difference in training time with vanilla ArcFace for R100?**
>
> **A7:** Thanks for your comment.
>
> Because most of the comparison methods are experiments done on the R100 framework. Therefore, in our paper, we mainly report the results of our TopoFR model on the R100 framework. In addition, we also provide the results of TopoFR model on the R50 and R200 frameworks.
>
> In Table 7, we only adopt the R50 framework as an example to compare the training time of our model with the vanilla ArcFace model because the R50 framework has the shortest training time.
>
> According to your advice, we have added the training time comparison results of the model on the R100 framework in Table 7 of the revised version. Please refer to the revised paper.

---

### Official Review · Reviewer_nN2M · 2023-11-01

**Soundness:** 4 excellent
**Presentation:** 3 good
**Contribution:** 4 excellent
**Rating:** 6
**Confidence:** 3

**Summary:**

This paper in face recognition (FR) proposes a topology-based approach for exploiting the  structural information present in face images.  The paper proposes perturbation-guided  topological structure alignment  (PTSA) for aligning the structural information in the input face images and latent feature spaces. The paper combines PTSA with a  hard sample mining strategy called structure damage estimation (SDE). Authors report experimental results on state of the art face image datasets (LFW, CFP-FP, AGEDB-30, IJB-C) with training on MS1MV2 and Glint360K.  An appendix containing many additional results is provided and both code and data are made available.

**Strengths:**

The major strength of the paper is the novel approach introduced. The PTSA method based on persistent homology (PH) appears to be novel in face recognition (FR) research and seems to outperform state of the art FR methods on pretty large FR datasets.

The experimental results presented are impressive. They not only show superior verification performance over state of the art FR methods, but authors seem to have conducted a pretty thorough experimental study and report a variety of results in the main paper and in the appendix.

**Weaknesses:**

The main weakness of this paper is that it is difficult to follow. Authors use many acronyms and technical phrases in early sections whereas those concepts are not explained until later sections. For example, Fig. 1 has  "death" and "birth" along the axes and refers to  "j-th dimension homologies" which are not defined.  Also, how do these figures in Fig. 1 indicate that there are high-dimensional holes as the amount of data increases? My estimation is that most readers will find this paper a hard read.

The paper also suffers from minor English language deficiencies, but these can be corrected during the revision.

**Questions:**

1. The paper refers to point clouds. Can you state clearly whether the face images are those taken with a normal RGB camera or whether these are point clouds from a LIDAR or something else.

2. Fig. 1:  What are the "death" and "birth" and what are j-th dimension homologies? How do these figures indicate that there are high-dimensional holes as the amount of data increases?

**Details Of Ethics Concerns:**

No ethics concerns.

---

> ### Author Response · Authors · 2023-11-22
> **Response to Reviewer nN2M**
>
> **Q1: The paper refers to point clouds. Can you state clearly whether the face images are those taken with a normal RGB camera or whether these are point clouds from a LIDAR or something else.**
>
> **A1:** Thanks for your valuable comment. All the face images are captured by RGB cameras. Actually, in our paper, we consider each face image or face feature as a data point in a high-dimensional space. As a result, a mini-batch of face images constitutes a point cloud in the space.
>
> **Q2: Fig. 1: What are the "death" and "birth" and what are j-th dimension homologies? How do these figures indicate that there are high-dimensional holes as the amount of data increases?**
>
> **A2:** Thanks for reminding us these issues. We have provided explanations for these concepts in Section 3 of the original manuscript, but perhaps they were not sufficiently comprehensive. Based on your suggestion, we provide the following specific explanations and definitions.
>
> **1) Birth and Death:**  Actually, the persistence diagram (PD) is a multi-set of points $(b, d)$ in the Cartesian plan $\mathbb{R}^{2}$, which encodes the birth time $b$ and death time $d$ information of each topological feature.
> Specially, each persistent point $(b, d)$ corresponds to a topological feature that emerges at scale $b$ and disappears at scale $d$. In other words, the topological feature is born at scale $b$ and dies at scale $d$.
>
> **2) About $j$-th dimension Homology:**  Homology is an algebraic structure that analyzes the topological features of a simplicial complex in different dimension $j$, including connected components $(H_{0})$, cycles/loops $(H_{1})$, voids/cavities $(H_{2})$, and higher-dimensional features $(H_{j},j\ge 3)$. By tracking the changes in topological features $(H_{j})$ across different dimensions $j$ as the scale parameter $\rho$ increases, we can obtain the multi-scale topological information of the underlying space [1].
>
> **3) About Figure 1:**  In topology theory, if the number of high-dimensional holes in the space is more, then the underlying topological structure of the space is more complex. Persistence diagram (PD) is a mathematical tool to describe the topological structure of space, where the $j$-th dimension homology $H_{j}$ in PD represents the $j$-th dimension hole in space. As shown in Figures 1(a)-1(d), as the amount of face data increases, the persistence diagram
> contains more and more high-dimensional homology (e.g., $H_{3}$ and $H_{4}$), which indicates that the input space contains more and more high-dimensional holes. Therefore, this phenomenon also shows that the topological structure of
> the input space is becoming more and more complex.
>
> Following your valuable advice, we have added the detailed description of Figure 1 in the revised version to better assist readers in understanding it. The modified sections are highlighted in blue.
>
> Ref. [1] Persistence images: A stable vector representation of persistent homology. JMLR 2017.
>
> **Q3: The paper also suffers from minor English language deficiencies, but these can be corrected during the revision.**
>
> **A3:** Thanks for taking the time to read the manuscript and reminding us of this issue. We will definitely fix this issue in the final version.

---

### Official Review · Reviewer_neNy · 2023-11-01

**Soundness:** 3 good
**Presentation:** 3 good
**Contribution:** 2 fair
**Rating:** 5
**Confidence:** 4

**Summary:**

In this paper, the authors propose a TopoFR approach for face recognition task.
The proposed TopoFR consists of two main components:
(1) a topological structure alignment strategy, namely PTSA, which adopts persistent homology to align the structure of input and latent spaces.
(2) A hard sample mining strategy, namely Structure Damage Estimation (SDE), and a structure damage score (SDS) to detect and prioritize the learning process of these hard examples.
TopoFR is validated on several FR benchmarks and shows its advantages in comparison to previous works.

**Strengths:**

- The paper is well-motivated.
- The idea of structural alignment is interesting.
- Experimental results show improvements in comparison to other baselines.
- Ablation study shows the contributions of each component.

**Weaknesses:**

There are some concerns on the design and experimental results.

1. Novelty: While I acknowledge the motivation of TopoFR approach, the novelty of its components is limited. For example, Diverse Data Augmentation (DDA) consists of common augmentation operators such as GaussianBlur, Grayscale, ColorJitter and Random Erasing (Zhong et al., 2020).
Moreover, Invariant Structure Alignments (ISA) is also adopted from previous work (i.e, Moor et al (2020).

Although it is ok for adopting previous works as building blocks, the novelty of the approach should be further emphasized. Otherwise, the paper becomes an incremental work.

2. Is the pairwise distance computed in Pixel Space (i.e. /matcal{X}) robust enough to estimate the topology? If this is sensitive, it should not be used to guide the learning process of the latent space.

3. In order to produce a good topology for learning process, how many samples should be used for each mini-batch?

4. How is the computational cost of the Structure Damage Estimation process during training?

5. What is the performance of the framework if only Focal Loss is used without SDS ?

6. The results on Figure 5 seems to be a bit tricky as TopoFR is trained with topological structure discrepancy metric. Adopting this metric for comparison will, of course, provide higher performance.

7. While large-scale model already gives high recognition results, the authors should adopt some more light-weight backbones. By this way, the contributions of the proposed components can be further emphasized.

**Questions:**

Please address the concerns in the Weakness section.

---

> ### Author Response · Authors · 2023-11-20
> **Response to Reviewer neNy Q1-Q5**
>
> **Q1: Novelty: While I acknowledge the motivation of TopoFR approach, the novelty of its components is limited. For example, DDA consists of common augmentation operators. Moreover, ISA is also adopted from previous work [1].**
>
> **A1:** Thanks for your valuable advice.
>
> Ref.[1] directly utilizes persistent homology to align the input and latent spaces of autoencoder. Unlike Ref.[1], in FR task, we find that directly aligning the structures of the input space and the latent space may cause the model to suffer "structure collapse phenomenon" (as stated in Section 1). Therefore, to address the structure collapse problem, we employ DDA mechanism to increase the latent space's structure diversity.
>
> Furthermore, it's worth mentioning that, in FR, most existing works do not include any augmentation operations, as this would introduce more unidentifiable face images, which generally does not bring benefit to the performance and even hurts the FR model's generalization ability, as stated and verified in Ref.[2,3]. Therefore, in this paper, we **do not employ these data augmentation strategies, such as GaussianBlur, Grayscale, ColorJitter and Random Erasing, to simply augment data scale.** Instead, we adopt DDA is to increase the latent space's structure diversity in order to effectively address the structure collapse problem. (More qualitative and quantitative analysis can be found in Section 1, Figure 2, Section 4.1, and Table 3.)
>
> Ref.[1] Topological autoencoders. ICML 2020.
>
> Ref.[2] Adaface: Quality adaptive margin for face recognition. CVPR 2022.
>
> Ref.[3] Towards universal representation learning for deep face recognition. CVPR 2020.
>
> **Q2: Is the pairwise distance computed in Pixel Space (i.e., $\mathcal{X}$) robust enough to estimate the topology? If this is sensitive, it should not be used to guide the learning process of the latent space.**
>
> **A2:** In the pixel space, we first flatten the face images into vector features and then calculate their distance matrix in order to construct Vietoris-Rips complex. And in the pixel space, there is no information loss on face features. Therefore, the computed topology is robust enough and can effectively guide the learning of the latent space structure.
>
> **Q3: In order to produce a good topology for learning process, how many samples should be used for each mini-batch?**
>
> **A3:** In our experiments, we adopt 4 A100 GPUs to train the TopoFR model, and a mini-batch of 128 face images is assigned for each GPU, as stated in section A.1 of Appendix. According your suggestion, we conduct several experiments to investigate the model's performance under different batch size, as shown in the Table R1.
> We find that increasing the batch size does not lead to a significant improvement in model's accuracy. Additionally, larger batch size imposes a greater workload on the GPU. Therefore, to strike a balance between model's accuracy and and GPU computational load, we choose to set the batch size to 128 in our model.
>
> **Table R1: Verification accuracy (%) on IJB-C.**
> | Training Data | Method | Batch Size | IJB-C(1e-4) | IJB-C(1e-5) |
> | ------ | ------ | ------ | ------ | ------ |
> |  | R100,TopoFR| 128 | **96.95** | **95.23** |
> | MS1MV2 | R100,TopoFR| 256 | 96.91 | 95.22 |
> | | R100,TopoFR| 512 | 96.93 | 95.20 |
>
> **Q4: How is the computational cost of the Structure Damage Estimation process during training?**
>
> **A4:** We have provided detailed training time analysis, please refer to the experiment named **"5) Training Time" and Table 7** in Appendix. The results presented in Table 7 demonstrate that introducing the Structure Damage Estimation (SDE) strategy leads to a significant performance gain with only a small increase in training time (i.e., 0.2s / 100 steps). Therefore, the addition of SDE does not bring too much computational burden.
>
> **Q5: What is the performance of the framework if only Focal Loss is used without SDS ?**
>
> **A5:** The ablation study results gathered in **Table 3** of the original manuscript demonstrate the model's performance when only using Focal loss. Variant TopoFR-F simply adopt Focal loss to re-weight sample, and achieve an accuracy of 96.73\% on "TAR@FAR=1e-4" of the IJB-C benchmark. Furthermore, the results in Table 3 also indicate that the prediction uncertainty $w_{1}$ is complementary to the Focal loss $w_{2}$ in mining hard samples.

---

> > ### Author Response · Authors · 2023-11-20
> > **Response to Reviewer neNy Q6-Q7**
> >
> > **Q6: The results on Figure 5 seems to be a bit tricky as TopoFR is trained with topological structure discrepancy metric. Adopting this metric for comparison will, of course, provide higher performance.**
> >
> > **A6:** Thanks for your comment.
> > As shown in Figure 2(d), we find that directly using persistent homology to align the topological structures of the input and latent space causes the model still leads to significant discrepancy between two spaces when evaluating on IJB-C benchmark, which indicates that the latent space fails to preserve the structure information of input space accurately. The occurrence of this issue is due to the model encountering the "structure collapse phenomenon", as stated in Section 1. It is worth noting that our PTSA strategy effectively alleviates the structure collapse phenomenon, resulting in smaller structure discrepancy during evaluation. The comparative results in Figure 2 clearly demonstrate the advantage of our method in reducing structure discrepancy.
> >
> > **Q7: While large-scale model already gives high recognition results, the authors should adopt some more light-weight backbones. By this way, the contributions of the proposed components can be further emphasized.**
> >
> > **A7:** Thanks for reminding us of this issue. Following your valuable suggestion, we add some experiments based on MobileFaceNet-0.45G [4], as shown in the Table R2.
> > We can observe that with the help of both SDE and PTSA strategies, the MobileFaceNet-0.45G model can achieve higher recognition accuracy, which demonstrates the effectiveness of our PTSA and SDE strategies.
> >
> > **Table R2: Verification accuracy (%) on IJB-C. MobileFaceNet refer to the MobileFaceNet-0.45G backbone.**
> > | Training Data | Method | IJB-C(1e-4) | IJB-C(1e-5) |
> > | ------ | ------ | ------ | ------ |
> > | MS1MV2 | MobileFaceNet | 93.42 | 90.13 |
> > | MS1MV2 | MobileFaceNet + PTSA + SDE| **94.48** | **91.01** |
> > | Glint360K | MobileFaceNet | 94.86 | 92.49 |
> > | Glint360K | MobileFaceNet + PTSA + SDE| **95.79** | **93.35** |
> >
> > Ref.[4] Mobilefacenets: Efficient cnns for accurate real-time face verification on mobile devices. 2018 CCBR.

---

> > ### Comment · Reviewer_neNy · 2023-11-22
> >
> > Thank you the authors for the response to my concerns. While the authors have addressed most of my concerns, there are still a few points:
> >
> > Q2. Pixel space is very sensitive to some factors (i.e. noise, illumination, pose, etc.) while latent features tend to remove them in their representation for robustness. therefore, the direct topology on pixel space may not be robust enough to guide the learning process.
> >
> > Q6. I think Figure 5 presents a trivial result. This is because ArcFace itself was not trained with topological structure discrepancy metric. Meanwhile, TopoFR is trained and evaluated on that metric. As a result, using the same structure discrepancy metric (during training and testing), TopoFR will get higher results.

---

> > > ### Author Response · Authors · 2023-11-23
> > > **Response to Reviewer neNy Q2_New and Q6_New**
> > >
> > > **Q2 New: Pixel space is very sensitive to some factors (i.e. noise, illumination, pose, etc.) while latent features tend to remove them in their representation for robustness. therefore, the direct topology on pixel space may not be robust enough to guide the learning process.**
> > >
> > > **A2 New:** Thanks for your comment. We agree with your viewpoint, and it is possible for such issues to exist in real-world scenarios.
> > >
> > > 1) The dimension of face features in the pixel space is significantly higher than that of the features in the latent layer space. Therefore, we believe that this can effectively capture the topological structure information hidden in the large-scale face dataset. In addition, no matter how the face images changes, their internal topological structure will not change significantly.
> > >
> > > 2) Furthermore, it is worth mentioning that the two face datasets we used (i.e., MS1MV2 [1] and Glint360K [2]) have undergone multiple rounds of cleaning, so they contain almost no noisy face samples. Therefore, the topological structure of pixel space can be well constructed.
> > >
> > > 3) Moreover, the expected k-th Betti number [3] $\mathbb{E}[\beta_{k}^{VR}(r)]=c_{k}n(nr^{d})^{2k+1}$ ($n$: data size, $d$: data dimension, $c_{k}$: constant) also demonstrates that as long as the dimension of the data is sufficiently high, its underlying topological structure can be well constructed.
> > >
> > > Based on the analysis above, we believe that the computed topology on pixel space is robust enough and can effectively guide the learning of the latent space structure.
> > >
> > >
> > > Ref.[1] Arcface: Additive angular margin loss for deep face recognition CVPR 2019.
> > >
> > > Ref.[2] Partial fc: Training 10 million identities on a single machine. ICCV 2021.
> > >
> > > Ref.[3] Random geometric complexes. DCG 2011.
> > >
> > > **Q6 New: I think Figure 5 presents a trivial result. This is because ArcFace itself was not trained with topological structure discrepancy metric. Meanwhile, TopoFR is trained and evaluated on that metric. As a result, using the same structure discrepancy metric (during training and testing), TopoFR will get higher results.**
> > >
> > > **A2 New:** Thanks for your valuable advice. We have understood your point. According to your suggestion, we make 2 corresponding revisions:
> > >
> > > (1) “using the same structure discrepancy metric (during training and testing)":
> > >
> > > We have modified the Figure 5 and its corresponding analysis.
> > > In order to more properly indicate the superority of PTSA strategy, we compare the the topological structure discrepancy of TopoFR and its variant TopoFR-A. Note that TopoFR-A directly employs PH to align the topological structures of two spaces.
> > > Please refer to the **Figure 5** and the section **"Generalization Performance of PTSA"** in the revised version.
> > >
> > > (2) "This is because ArcFace itself was not trained with topological structure discrepancy metric.":
> > >
> > > In order to better show the generalization ability of our method, we further utilize the Bottleneck distance metric to compare the topological structure discrepancy between two spaces of vanilla ArcFace and our TopoFR models. Please refer to the **Figure 8** and the section **"Comparison of Structure Discrepancy"** in the revised appendix.
> > >
> > > We believe that the above experimental results are sufficient to demonstrate the superior generalization ability of our PTSA in preserving structure information.
> > >
> > >  If there are any other concerns, please feel free to let us know. Thanks again for your time and efforts!

---

> ### Author Response · Authors · 2023-11-22
> **Looking forward to the reply**
>
> Dear reviewer neNy:
>
> Thanks so much again for the time and effort in our work. According to the comments and concerns, we conduct the corresponding experiments and further discuss the related points.
>
> As the discussion period is nearing its end, please feel free to let us know if there are any other concerns. Thanks again for your time and efforts.

---

### Meta-Review · Area_Chair_dqrA · 2023-12-06

**Metareview:**

The paper proposes TopoFR, a topology-based face recognition approach comprising Perturbation-Guided Topological Structure Alignment (PTSA) and Structure Damage Estimation (SDE) for hard sample mining. PTSA uses persistent homology to align structural information, while SDE prioritizes challenging examples. However, it is still limited in novelty as reviewers argued. For example, Diverse Data Augmentation (DDA) consists of common augmentation operators such as GaussianBlur, Grayscale, ColorJitter and Random Erasing (Zhong et al., 2020). Moreover, Invariant Structure Alignments (ISA) is also adopted from previous work (i.e, Moor et al (2020). And The observation (a) is not novel, as it was theoretically investigated before [Kahle11,Bobrowski18], more on this later. Also, this paper may be difficult to follow. Authors use many acronyms and technical phrases in early sections whereas those concepts are not explained until later sections.

**Justification For Why Not Higher Score:**

As illustrated in the metareview, This paper is limited in novelty and presentation. It doesn't meet the requirement of the ICLR, so it can not get higher score.

**Justification For Why Not Lower Score:**

N/A

---

### Decision · Program_Chairs · 2024-01-16

Reject